# Functional Characteristics of Serum Anti-SARS-CoV-2 Antibodies against Delta and Omicron Variants after Vaccination with Sputnik V

**DOI:** 10.3390/v15061349

**Published:** 2023-06-10

**Authors:** Elizaveta I. Radion, Vladimir E. Mukhin, Alyona V. Kholodova, Ivan S. Vladimirov, Darya Y. Alsaeva, Anastasia S. Zhdanova, Natalya Y. Ulasova, Natalya V. Bulanova, Valentin V. Makarov, Anton A. Keskinov, Sergey M. Yudin

**Affiliations:** Federal State Budgetary Institution, Centre for Strategic Planning and Management of Biomedical Health Risks of the Federal Medical Biological Agency, Schukinskaya 5, Building 1, Moscow 123182, Russia

**Keywords:** SARS-CoV-2, vaccination, neutralizing antibodies, ADMP, Sputnik V

## Abstract

Anti-SARS-CoV-2 vaccination leads to the production of neutralizing as well as non-neutralizing antibodies. In the current study, we investigated the temporal dynamics of both sides of immunity after vaccination with two doses of Sputnik V against SARS-CoV-2 variants Wuhan-Hu-1 SARS-CoV-2 G614-variant (D614G), B.1.617.2 (Delta), and BA.1 (Omicron). First, we constructed a SARS-CoV-2 pseudovirus assay to assess the neutralization activity of vaccine sera. We show that serum neutralization activity against BA.1 compared to D614G is decreased by 8.16-, 11.05-, and 11.16- fold in 1, 4, and 6 months after vaccination, respectively. Moreover, previous vaccination did not increase serum neutralization activity against BA.1 in recovered patients. Next, we used the ADMP assay to evaluate the Fc-mediated function of vaccine-induced serum antibodies. Our results show that the antibody-dependent phagocytosis triggered by S-proteins of the D614G, B.1.617.2 and BA.1 variants did not differ significantly in vaccinated individuals. Moreover, the ADMP efficacy was retained over up to 6 months in vaccine sera. Our results demonstrate differences in the temporal dynamics of neutralizing and non-neutralizing antibody functions after vaccination with Sputnik V.

## 1. Introduction

The emergence of COVID-19 has led to the development of an unprecedented global pandemic, causing more than six million deaths worldwide. Since the beginning of the spread, SARS-CoV-2 has accumulated dozens of mutations, dividing into numerous variants with different frequency distributions [1]. These mutations are not uniformly distributed in the SARS-CoV-2 genome, with the greatest number of differences being observed in the S-protein [2]. Indeed, the S-protein acts as the main target of neutralizing antibodies and, hence, is under strong selection pressure. Many vaccines for SARS-CoV-2 have been developed over the past two years since the beginning of the pandemic, and most of them use the 2019-nCoV reference S-protein as the main antigen [3]. Anti-S-protein antibodies can neutralize viral binding to the ACE2 receptor (Angiotensin-converting enzyme 2) as well as mediating various effector links of the immune system, such as complement activation, antibody-dependent phagocytosis, and activation of natural killer cells [4]. These later functions are provided by non-neutralizing antibodies which are produced during the immune response to infection or vaccination.

In Russia, the Sputnik V vaccine (Gam-COVID-Vac) is widely used. The Sputnik V has shown high efficiency during the dominance of early SARS-CoV-2 variants [5,6,7,8,9,10]; however, its effectiveness against new SARS-CoV-2 variants demonstrates a decrease in neutralizing activity. Indeed, the neutralization activity of Sputnik V vaccine sera against B.1.351 (Beta), P.1 (Gamma), B.1.617, and B.A1 (Omicron) decreases by several times compared to the Wuhan-Hu-1 SARS-CoV-2 G614-variant (D614G) [11,12,13,14,15,16]. Moreover, previously published papers show a significant decrease of Omicron neutralization by BNT162b2, mRNA-1273, Ad26.COV2. S, and ChAdOx1 vaccine sera [17,18,19,20,21].

Despite the large number of published studies on evaluating the effectiveness of vaccine-driven immunity, most of them are focused on studying the neutralization activity of vaccine sera. Meanwhile, the non-neutralization activity of vaccine sera remains poorly assessed. This side of the immune response is an important aspect of immunity and affects survival and the course of the disease. Moreover, the non-neutralizing immune response might have different changing dynamics over time. For example, despite the reduction in the neutralizing activity, the Sputnik V vaccine remains effective against hospitalization and severe lung injury associated with SARS-CoV-2 Omicron [22,23], which probably might be explained by the retention of activity of the effector links of the immunity. In the current study, we performed a functional characterization of anti-SARS-CoV-2 antibodies against Delta and Omicron variants after vaccination with Sputnik V. We aimed to assess the differences in reduction in neutralizing and Fc-mediated activity of vaccine sera for up to 6 months against B.1.617.2 (Delta) and B.A1. Our results demonstrate interesting differences between these two closely related aspects of the immune response.

## 2. Materials and Methods

### 2.1. Ethics Approval

This study was approved by the ethical committee of the Federal State Budgetary Institution “Centre for Strategic Planning and Management of Biomedical Health Risks” (Protocol No. 5 from 01/04/2021).

### 2.2. Serum Samples and Study Design

In the vaccinated group, serum samples were collected from individuals who received two doses of Sputnik V vaccine and did not have confirmed SARS-CoV-2 infection at the time of sample collection (*n* = 36), as well were seronegative for the anti-N-protein-specific IgGs (Table 1). In the COVID-19-recovered group, serum samples were collected from individuals who had recovered from COVID-19 between December 2021 and March 2022 (*n* = 26). During the sample collection campaign, a PCR analysis was performed, which allowed to distinguish between the Delta and Omicron lines (AmpliTest, SARS-CoV-2 VOC v3, CV017, Russia). The individuals from COVID-19-recovered group had the following additional characteristics: 18 out of 26 individuals (69%) had been previously vaccinated with two doses of Sputnik V. A total of 9 out of 26 individuals (35%) have had previous COVID-19 infection according to their anamnesis data (Table 2). The samples were collected according to the standard procedure of vacuum-venipuncture into tubes containing coagulation activator and separating gel. The serum samples were separated after centrifugation at 3000 rpm for 10 min, transferred into new tubes, and stored at −80 °C until they were processed for laboratory assays.

### 2.3. Quantification of Serum Anti-SARS-CoV-2 S1, Anti-RBD, and Anti-N Antibody Levels

The quantification of anti-SARS-CoV-2 S1, anti-RBD, and anti-N antibody levels in serum samples were performed using the MILLIPLEX SARS-CoV-2 Antigen Panel 1 IgG kit. This panel is designed to measure antibodies using median fluorescent intensity (MFI). The Luminex assay uses distinct fluorescent microspheres, on which each S1, RBD (receptor-binding domain), and N protein is individually coupled and eventually mixed to create a multiplex set. Antibodies in the sample bind to protein-bound beads and are then detected by using anti-human IgG antibodies conjugated to phycoerythrin (PE).The samples were run on a Luminex FlexMap 3D instrument with the xPONENT 4.3 software. The data analysis was carried out with the software MILLIPLEX Analyst 5.1.

### 2.4. Plasmids

The coding sequences of ACE2 (NM_001371415.1) were cloned into pLVX-Puro (Clontech). The coding sequences of TMPRSS2 (NM_001135099.1) were cloned into pLVX-BleoR. D614G and B.1.617.2 (Delta) S-protein coding sequences (Figure 1A) were codon optimized and synthesized on BioXP5 Codex DNA, Inc.; BA.1 (Omicron) coding sequence was codon-optimized and synthesized by ShineGene Molecular Biotech, Inc. To increase the assembly efficiency of the pseudotyped lentiviral particles, 19 amino acids from each S-protein C-terminus were removed, after which the final sequences (Appendix A) were cloned into the pcDNA3.3-topo-TA vector.

### 2.5. Cell Lines

HEK293T-hACE2 cells were generated using lentiviral transduction of HEK293T cells with pLVX_hACE2_Puro bearing lentiviruses, followed by selection in a complete medium (DMEM (PanEco), 10% FBS, 1x penicillin-streptomycin (PanEco) with puromycin (InvivoGen) (1 µg/mL). HEK293T-hACE2-TMPRSS2 cells were generated by the lentiviral transduction of HEK293T-hACE2 cells with pLVX_TMPRSS_BleoR containing lentiviruses, followed by selection in a complete medium containing puromycin and bleomycin (Thermo Fisher Scientific) (5 µg/mL).

### 2.6. Assembly of Pseudotyped Lentiviral Particles

Pseudotyped lentiviral particles were generated by the co-transfection of HEK293T cells with psPAX2 (addgene #12260), pLV-PGK-GFP plasmids, and a plasmid carrying the corresponding S-protein sequence. 24 h after -transfection, the cultural media was changed to virus-producing medium (Opti-MEM (Gibco), 5% FBS (PanEco), and 1 mM sodium pyruvate (PanEco)). Next, after 24 h, the virus-containing supernatant was collected, filtered through a 0.45 μm polyester-sulfone filter, frozen, and stored at −80 °C. The resultant pseudotyped particles were titrated on HEK293T-hACE2-TMPRSS2 cells, and the infection percentage was determined by flow cytometry. The titers of the pseudotypes were equalized before the neutralization assay.

### 2.7. Neutralization Assay

We seeded 7 × 10^3^ target cells per well in a 96-well plate in 100 μL of complete medium 24 h prior to infection. Infection was carried out in a complete medium supplemented with polybrene (8 μg/mL). Immediately prior to infection, the serum samples were heat-inactivated at 56 °C for 30 min and then diluted in a range from 1/10 to 1/1280 of medium’s total volume. For the neutralization assay, 30 μL of virus-containing supernatants were incubated with 20 μL of diluted inactivated serum for 2 h at 37 °C, and then 50 μL of the complete medium was added. Next, the virus–serum mixtures were added to the cells, and the cells were incubated for 72 h at 37°C, 5% CO_2_. Cells were washed with PBS containing 5 mM EDTA and analyzed using a CytoFLEX LX Flow Cytometer (Beckman Coulter). The neutralization efficiency was calculated using the percentage of GFP-positive cells as a read out.

### 2.8. Antibody-Dependent Cellular Phagocytosis

An assessment of antibody-dependent monocyte-mediated phagocytosis (ADMP) was performed on the THP-1 cells. For this, the recombinant S-proteins of D614G, B.1.617.2, and BA.1 SARS-CoV-2 variants (Miltenyi Biotec) were biotinylated using a protein-biotinylating kit (Lumiprobe), followed by incubation for 12 h with fluorescent 1 µm neutravidin particles (Invitrogen, F8775) at 4 °C. Next, the resultant particles were incubated with heat-inactivated serum samples at 37 °C for 2 h. The THP-1 cells were maintained in RPMI medium (PanEco) supplemented with 10 mM L-Glutamine and 10% FBS (PanEco). A total of 25 × 10^4^ of the THP-1 cells were seeded in a 96-well plate in serum-free AIM-V medium (Gibco), and then the S-protein-covered fluorescent particles were added and incubated overnight at 37 °C, 5% CO_2_. Next, the cells were washed with PBS, fixated with 4% PFA, and analyzed using flow cytometry using the CytoFlex LX (Beckman Coulter). The antibody-dependent phagocytic activity index was calculated as follows: (% of cells containing particles × MFI (mean fluorescent intensity) of cells containing particles)/10^4^ [24,25].

### 2.9. Statistical Analysis

The statistical analysis was performed using the GraphPad Prism 9.4.1 application package. The ID50 parameter (the serum dilution value required to reduce the proportion of GFP-positive target cells by 50%) was calculated using a non-linear regression model (asymmetric sigmoidal 5PL) for each serum sample. For quantitative traits, unrelated groups were compared using non-parametric Mann–Whitney (U-test) and Kruskal–Wallis (K–W ANOVA) tests. The dependence analysis was conducted using the Spearman rank correlation method. The differences were considered statistically significant at the significance level *p* < 0.05.

## 3. Results

### 3.1. Dynamics of IgG Responses and Neutralization after Sputnik V Vaccination

In the beginning of our study, we addressed the anti-SARS-CoV-2 S1, anti-RBD, and anti-N antibody levels in Sputnik V-vaccinated serum samples. For this, we performed the quantification of antibody levels on a FlexMap 3D Luminex analyzer using the MILLIPLEX SARS-CoV-2 Antigen Panel 1 IgG kit. We found sustainable antibody production of anti-SARS-CoV-2 S1 and anti-RBD antibodies in 98.5% (61 of 62) of vaccine serum samples; meanwhile, none of the serum samples demonstrated any significant anti-N antibody levels. As expected, we observed a decrease in anti-S1 and anti-RBD antibody levels over time after vaccination, with a significant reduction 6 months after vaccination (Figure 1A).

Next, to assess anti-SARS-CoV-2 neutralizing antibody levels, we analyzed the neutralizing activity of serum samples using an S-protein-pseudotyped lentiviral assay (Appendix A). To develop the S-protein-pseudotyped lentiviral assay, we used plasmid constructs, encoding for S-proteins of the D614G, B.1.617.2, and BA.1 SARS-CoV-2 variants (Appendix A). To increase the assembly efficiency of the pseudotyped lentiviral particles, we removed 19 amino acids from each S-protein C-terminus. These amino acids form the presumptive ERR (endoplasmic reticulum retention) motif, also known as KxHxx. It was previously shown that this motif is necessary for the incorporation of S-proteins into the ER-Golgi intermediate compartment (ERGIC), which is one of the stages of SARS-CoV-2 virion maturation [26,27].

We aimed to characterize the changing dynamics of serum neutralizing activity against D614G, B.1.617.2, and BA.1 in Sputnik V-vaccinated individuals over time after vaccination. For this, we compared the serum samples from Sputnik V-vaccinated individuals collected 1 month (*n* = 36), 1–4 months (*n* = 16), and 4-6 months (*n* = 10) after the second vaccine dose (Figure 1B). We found that 1 month after vaccination, the serum neutralizing activity against the D614G and B.1.617.2 SARS-CoV-2 variants was detected in 97% (35 of 36) and 87% (31 of 36) patients, respectively. At the same time, only 28% (10 of 36) of the serum samples neutralized BA.1; moreover, the ID50 values for BA.1 were 8.16 times lower than for the original D614G variant (*p* < 0.0001). Then we observed a decrease in ID50 values for all studied SARS-CoV-2 variants 4 and 6 months after vaccination. Interestingly, 70% (7 of 10) and 60% (6 of 10) of all studied serum samples neutralized D614G and B.1.617.2, respectively; however, only 20% (2 of 10) of serum samples were able to neutralize the BA.1 variant. In addition, we observed that 6 months after vaccination, ID50 values (compared to the original ID50 for D614G) decreased by 3.67-, 4.29-, and 11.6-fold for SARS-CoV-2 variants D614G, B.1.617.2, and BA.1, respectively. Our results are in concordance with previously published works. For example, in the work of Lapa et al., 2022 [28], the authors assessed the reduction in neutralization activity of Sputnik V vaccine sera against Omicron (B.1.1.529). The study demonstrates an 8.8-fold decrease in serum neutralizing activity against B.1.1.529 compared to the Wuhan variant. Interestingly, the reduction in our study is more substantial, which might be explained by differences in the methodology and sample size.

### 3.2. ADMP Retention over Time after Vaccination with Sputnik V

Next, to assess another function of antibodies, we conducted an antibody-dependent monocyte-mediated phagocytosis (ADMP) assay. In contrast to neutralization, which is mediated by the binding of antibodyFab-fragments to RBD, ADMP is mostly mediated by Fc-fragments of antibodies [29]. First, we aimed to assess possible distinctions between the ADMP-triggering efficacies of Sputnik V vaccine serum samples (*n* = 26) against all of the S-protein variants studied. Our ADMP assay demonstrated no statistically significant differences between all of the S-protein variants studied (Figure 1C). Next, we aimed to track the changing dynamics of ADMP over time after vaccination. For this, we compared ADMP responses for the serum samples from Sputnik V-vaccinated individuals collected 1 month (*n* = 12), 1–4 months (*n* = 8), and 4–6 months (*n* = 6) after the second vaccine dose. In contrast to the neutralization activity dynamics, we observed no significant variation in different time points after vaccination (Figure 1D). These results are in agreement with previously published works. A recent study on ADMP dynamics after SARS-CoV-2 infection also demonstrates stability of ADMP responses for over 8 months after infection [30].

Finally, we assessed the interconnection between the total anti-S1 and anti-RBD antibody levels and both neutralizing activity and ADMP efficacy against the studied variants. The Spearman rank–order correlation analysis demonstrated a significant connection between anti-S1 and anti-RBD antibody levels and neutralizing activity against the D614G and B.1.617.2 variants, but to a lesser extent against the BA.1 variant (Figure 1E).

Interestingly, in the work by Vangeti et al. 2022, the authors observed ADMP correlation with only anti-S1 and anti-S2 antibody level dynamics, whereas anti-RBD IgG levels demonstrated no significant correlation. Such a discrepancy with our correlation results may be explained by distinct methodologies for the assessment of antibody levels or differences in immune response after infection and vaccination.

**Figure 1 viruses-15-01349-f001:**
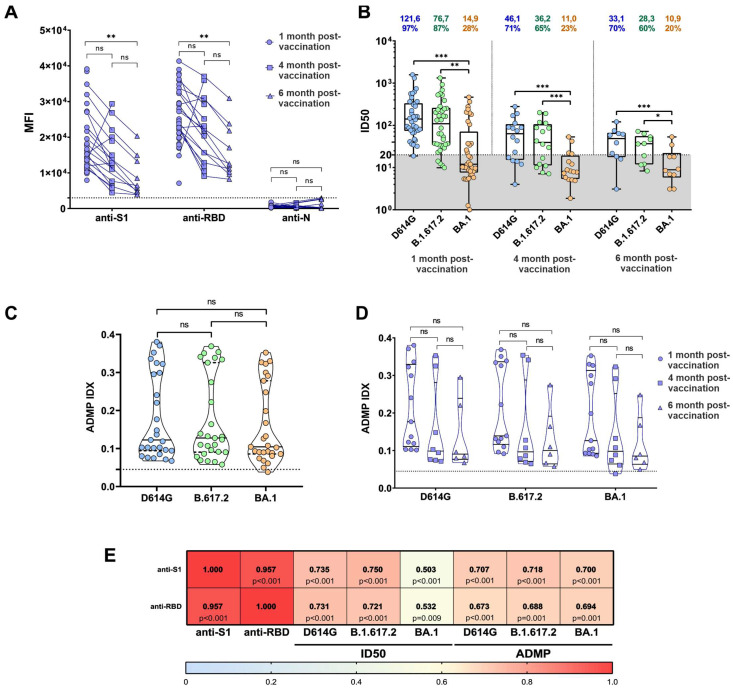
Functional characteristics of anti-SARS-CoV-2 serum antibodies after Sputnik V vaccination. (**A**) Quantitative assessment of the anti-S1, anti-RBD, and anti-N antibody production in serum samples from Sputnik V-vaccinated study participants. The Y-axis represents the mean fluorescence intensities (MFI). The dotted line represents the cutoff value for the FlexMap results. *p*-values were determined by using the Kruskal–Wallis test. (**B**) Neutralization of pseudotyped lentiviruses by the Sputnik V vaccine sera in time after vaccination. The ID50 and neutralizing serum percentages are shown above the plot. The gray zone represents the cutoff value. The given cutoff value means that 95% of negative sera samples have ID50 parameters equal to 20 or lower. *p*-values were determined by using the Kruskal–Wallis test. (**C**) Graphical representation of ADMP efficacy against SARS-CoV-2 variants D614G, B.1.617.2, and BA.1. No statistically significant differences were found. The dotted line represents the cutoff value for the ADMP index. *p*-values were determined by using the Kruskal–Wallis test. (**D**) ADMP efficacy in time after vaccination against SARS-CoV-2 variants D614G, B.1.617.2, and BA.1. No statistically significant differences were found. *p*-values were determined by using the Kruskal–Wallis test. The dotted line represents the cutoff value for the ADMP index. (**E**) The interconnection between the total anti-S1 and anti-RBD antibody levels with both neutralizing activity and ADMP efficacy against D614G, B.1.617.2, and BA.1; *p*-values are shown in each box. The heat map represents the value of Spearman rank correlation coefficient (*: *p* < 0.05, ** : *p* < 0.01, *** : *p* < 0.001, ns: no significance).

### 3.3. Previous Sputnik V Vaccination Does Not Enhance the Serum Neutralization Activity against BA.1 for Patients Who Have Recovered from SARS-CoV-2 Omicron

Previous studies show that SARS-CoV-2 infection induces the production of neutralizing antibodies in most infected patients [31,32,33]. We studied the effect of prior vaccination with Sputnik V on the production of IgG, as well as serum neutralizing activity, after SARS-CoV-2 Omicron infection. For this, we collected convalescent serum samples from COVID-19-recovered patients (*n* = 26), among which 18 patients (69%) had been previously vaccinated with two doses of Sputnik V. The median post-vaccination time was 171 days (12–299 days). First, we assessed the anti-S1, anti-RBD and anti-N antibody levels in serum samples using the FlexMap 3D Luminex analyzer by the MILLIPLEX SARS-CoV-2 Antigen Panel 1 IgG kit. We found sustainable production of anti-S1 and anti-RBD antibodies in all serum samples; meanwhile, anti-N antibodies were detected in 83.3% of the convalescent patients (15 of 18) (Figure 2A). Next, we analyzed the neutralization activity of the collected serum samples against D614G, B.1.617.2, and BA.1. Our results show that the ID50 for D614G was higher for recovered Sputnik V-vaccinated Omicron patients compared to recovered unvaccinated participants (*p* < 0.05) (Figure 2B). At the same time, we found no statistically significant differences between the groups for ID50 BA.1 (Figure 1D), while the ID50 for D614G and B.1.617.2 was lower than for BA.1 in the group of unvaccinated recovered patients (*p* < 0.05).

## 4. Discussion

In this study, we have conducted functional assessments of serum anti-SARS-CoV-2 antibodies against Delta and Omicron variants after vaccination with Sputnik V. Initially, we addressed the dynamics and longevity of antibody response after Sputnik V vaccination. We have shown a significant decrease in anti-S1 and anti-RBD antibody levels 6 months after Sputnik V vaccination. Then we conducted a functional analysis of serum anti-SARS-CoV-2 antibodies against D614G, B.1.617.2, and BA.1 variants. For this, we first assessed the neutralization activity of SARS-CoV-2 BA.1 in the sera compared to D614G and B.1.617.2 in Sputnik V-vaccinated patients. It was previously shown that serum neutralizing activity is determined by the number of neutralizing antibodies produced in response to a previous infection or vaccination [34]. In concordance with that, our data on the antibody quantifications show a strong correlation between anti-S1 and anti-RBD antibody levels and serum neutralizing activity against the D614G and B.1.617.2 variants; however, for the Omicron BA.1 variant, we observed only a weak correlation. The neutralizing activity of antibodies is determined by their ability to bind to specific epitopes of the antigen [35]. In the case of Sputnik V vaccination, the antigen is the S-protein of the 2019-nCoV reference variant, which differs greatly from the S-protein sequence of current SARS-CoV-2 variants. The most pronounced differences are in the S-protein of Omicron variants, which carry more than three dozen non-synonymous substitutions, with 15 of them located within RBD [36,37]. Interaction of RBD with ACE2 is necessary for the virus to enter the target cells [38,39,40]. It has been previously shown that blocking this interaction with RBD-specific antibodies neutralizes the virus, preventing its penetration into cells [41]. Presumably, in case of BA.1 SARS-CoV-2 variant, the weak affinity of the Sputnik V vaccine neutralizing anti-RBD antibodies does not completely prevent their binding to ACE2, which is not enough to neutralize the virus.A recent study on the Sputnik V vaccine’s antibody neutralization activity using a microneutralization plaque assay demonstrated an 8.1-fold decrease in neutralization for the B.1.1.529 variant compared to D614G [28]. In the present study, we carried out an independent assessment of Sputnik V-vaccine serum neutralizing activity on a well-characterized sample of a larger size, which complements the previously published results.

The changing dynamics of neutralizing activity of convalescent serum with time have been demonstrated in many works [34,42,43,44]. It has been shown that the serum neutralizing activity in recovered patients can change according to several scenarios: a rapid or slow decrease, stable persistence, or an increase in neutralizing activity due to a delayed start of antibody production [38]. In our work, we assessed the duration of neutralizing antibody persistence in Sputnik V vaccine sera within 6 months after vaccination. We have shown that a pronounced decrease in vaccine serum neutralizing activity against D614G and B.1.617.2 occurs in the first 4 months after vaccination (by 26 and 22%, respectively), while in the next 2 months, the neutralizing activity against them drops slightly. A similar pattern was observed for the vaccine serum neutralizing activity against the Omicron BA.1 variant. Earlier, in previous works by other authors, immunity duration models were built based on the level of neutralizing antibodies [40]. Such models are especially important for planning the revaccination schedule.

Apart from viral receptor recognition and cell entry blockage, IgG may trigger antibody-dependent monocyte-mediated phagocytosis (ADMP). ADMP promotes the removal of the virus and the virus-infected cells, serving as an additional mechanism of antiviral defense. In contrast to neutralization mechanisms, the ADMP triggering function is mediated by Fc-fragments of antibodies. In the process of ADMP, the antibody–pathogen immune complexes bind to FcR located on the phagocytes via Fc-fragments of the antibodies. Next, the ingested immune complexes are directed to lysosomes, where they are degraded and then are loaded onto MHC molecules for presentation and priming of T cells [45]. ADMP plays important role in pathogenesis of various diseases. For example, in a mouse model of SARS-CoV infection, phagocytes contributed to the antibody-mediated elimination of pulmonary-infected SARS coronavirus [46]. For SARS-CoV-2, the impaired ADMP function was linked to infection-related mortality [47]. In the frame of our work, we assessed the ADMP-triggering efficacy of Sputnik V vaccine serum samples collected at different time points after vaccination. We demonstrated that the ADMP efficacy for the D614G, B.1.617.2, and BA.1 variants did not differ significantly in vaccinated individuals. Moreover, the ADMP efficacy demonstrated no significant reduction over the 6 months after vaccination. Such ADMP stability does not coincide with a decrease in both antibody levels and neutralization efficiency. This discrepancy might be explained by the broad functional antibody spectrum produced in the frame of the humoral immune response. Upon infection, the immune response results in the production of both neutralizing and non-neutralizing antibodies. Non-neutralizing anti-SARS-CoV-2 antibodies do not prevent viral cell entry; however, they can mediate the activation of immune effector cells through interaction between IgG Fc-fragments and FcR. These Fc-fragments are constant and do not correlate with antibody specificity to a particular antigen. In the case of SARS-CoV-2, the antibody specificity to a particular SARS-CoV-2 variant seems to play no major role in the ADMP response. In addition, the discrepancy between neutralization and ADMP efficacy dynamics might be explained by different bottom lines for the antibody levels required for each function. Since neutralization is antigen-specificity-dependent, the bottom line of antibody levels required for this process might be higher than for Fc-mediated functions. Thus, the remaining anti-SARS-CoV-2 antibody levels might be sufficient for Fc-mediated functions, but not sufficient for neutralization. The problem of the bottom line of antibody levels necessary for Fc- and Fab- mediated function requires further studies and goes beyond the scope of the current study.

Previous studies by other authors have shown that booster vaccination with mRNA vaccines significantly increases the serum neutralization activity of BA.1/B.1.1.529 [48,49]. Our study did not consider the effect of booster vaccinations on neutralization, but we investigated the effect of previous Sputnik V–vaccination on the convalescent serum neutralizing activity of recovered Omicron patients. We found no effect of previous vaccination on BA.1 neutralization, while for D614G, the neutralization in unvaccinated recovered individuals was lower than for vaccinated recovered patients. In the case of BA.1 infections, the antibodies produced appear to have a reduced affinity for the RBD of the D614G variant. It can be assumed that these antibodies are not capable of effective recognition of the D614G S-protein, and, thus, they weakly neutralize D614G.

The vaccination strategy for pandemic containment has shown its effectiveness during the dominance of early SARS-CoV-2 variants. Now, in times of the emerging of new variants with multiple substitutions within the S-protein, the major antigen of many vaccines, the problem of the previous vaccination campaign’s effectiveness stands out. It is necessary to develop new vaccines that are more resistant to new SARS-CoV-2 variants. The main immunodominant component of such vaccines may be a more conserved antigen showing fewer fixed mutations [50,51,52,53,54]. One such alternative strategy might be the use of more conserved SARS-CoV-2 epitopes as immunodominant components. Besides S-protein, nucleocapsid (N) protein is the second most conservative SARS-CoV-2 protein [55]. N protein is conserved among various coronaviruses and causes the production of cross-reacting antibodies [56]. However, in neutralization experiments, anti-N antibodies did not demonstrate significant neutralizing activity [57,58]. This imposes restrictions on the use of N-protein as a vaccine antigen. Alternatively, recent studies propose another strategy for anti-SARS-CoV-2 vaccine development. In recent work from Zhao et al., the authors suggested the use of a new pan-vaccine antigen (S_pan_). The S_pan_ was developed by analyzing more than 2000 sequences of SARS-CoV-2 S-protein over the last few years. The resulted S_pan_ sequence contains high-frequency residues at given positions, which ensures the formation of anti-SARS-CoV-2 antibodies that can neutralize many variants of the virus [59]. This strategy might be the most promising since it allows for competition with the antigenic drift caused by rapid viral evolution.

## 5. Study Strengths and Limitations

The main strength of this study is the use of a high-throughput quantitative method of assessing neutralizing antibody levels without the use of the infectious virus. On the other hand, the use of pseudotyped lentiviruses instead of native viral particles may impose limitations on the results obtained. Since the surface of the real virus is a complex system comprising additional viral proteins, the extent of the serum neutralization capability demonstrated using the pseudotyped lentiviral system may differ from one assessed by native viral particles. This possible difference requires additional studies and is not included in the scope of this study.

Also, another strength of the current study is the fixed time schedule for the collection of serum samples from Sputnik V-vaccinated individuals. However, the small sample size, lack of randomization, and unavailability of a placebo group can be considered limitations of the current study.

## Figures and Tables

**Figure 2 viruses-15-01349-f002:**
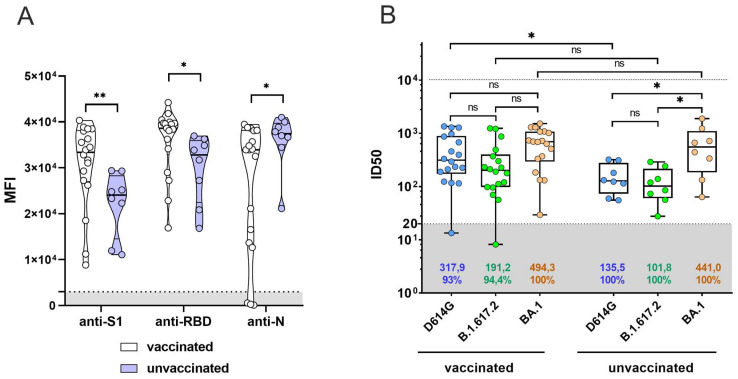
IgG production and neutralization of pseudotyped lentiviruses by COVID-19 convalescent sera. (**A**) Quantitative assessment of the of anti-S1, anti-RBD, and anti-N antibody production in recovered SARS-CoV-2 patients’ serum samples. The Y-axis represents the mean fluorescence intensities (MFI). The gray zone represents the cutoff value for the FlexMap results. *p*-values were determined by using the Mann–Whitney test. (**B**) Neutralization of pseudotyped lentiviruses by convalescent sera. The ID50 and neutralizing serum percentages are shown below the plot. *p*-values for comparison between vaccinated and unvaccinated group were determined by using the Mann–Whitney test, and *p*-values for the comparison between SARS-CoV-2 variants inside each group were determined by using the Kruskal–Wallis test. The cutoff value is shown as the gray zone (*: *p* < 0.05, ** : *p* < 0.01, ns: no significance).

**Table 1 viruses-15-01349-t001:** The Sputnik V study group summary data.

Serum Samples from Sputnik V-Vaccinated Individuals (*n* = 62)			
	1 Month after Vaccination (*n* = 36)	4 Months after Vaccination (*n* = 16)	6 Months after Vaccination (*n* = 10)
Gender, n (%) male female	17 (47%) 19 (53%)	7 (44%) 9 (56%)	4 (40%) 6 (60%)
Age, years Median (min–max)	46 (27–79)	43 (27–79)	50 (29–76)
Period of inclusion into the study	May–August 2021		
Interval between sample collection and vaccination, days Median (min–max)	42 (21–42)	120 (120–120)	180 (180–180)
Previously had confirmed cases of COVID-19, *n* (%)	0	0	0

**Table 2 viruses-15-01349-t002:** The COVID-19-recovered study group summary data.

Serum Samples from SARS-CoV-2-Recovered Individuals (*n* = 26)	
Gender, *n* (%) Male Female	11 (42%) 15 (58%)
Age, years Median (min–max)	46 (18–72)
Period of inclusion into the study	December 2021–March 2022
Interval between sample collection and clinical onset, days Median (min–max)	36 (20–67)
Previously vaccinated, n (%)	18 (69%)
Interval between clinical onset and previous vaccination, days Median (min–max)	171 (12–299)
Previously had confirmed cases of COVID-19, n (%)	9 (35%)
Interval between clinical onset and previous confirmed COVID-19 case, days Median (min–max)	269 (94–457)

## Data Availability

The data that support the findings of this study will be shared upon reasonable request to the corresponding author.

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
