# Peer review of "Functional Characteristics of Serum Anti-SARS-CoV-2 Antibodies against Delta and Omicron Variants after Vaccination with Sputnik V"

_viruses, 2023, doi:10.3390/v15061349_

Round 1
Reviewer 1 Report
The revised manuscript entitled “Functional characteristics of serum anti-SARS-CoV-2 antibodies against Delta and Omicron variants after vaccination with Sputnik V” presents data of the original research aimed to assessment and comparison of the neutralizing activity in the Sputnik V vaccinated individuals against two SARS-CoV-2 variants of concern in Moscow, Russia. The authors used modern methods for detection of the coronavirus specific immunoglobulins, neutralizing activity of genetically engineered pesudoviruses and antibody-dependent cellular phagocytosis. The manuscript is well written and represents interesting results. However, there are several features which, unfortunately, prevent the publication of the paper in its present form.
The major concern is as next. The study in its present form focuses on the demonstration of the “reduced neutralizing activity of Sputnik V vaccine sera against modern SARS-CoV-2 variants - B.1.617.2 and B.1.1.529.1”. However, up to date this issue is thoroughly studied. The virus-neutralizing activity against Delta and Omicron in sera and plasma of Sputnik V and Sputnik Light vaccinated individuals has been assessed in a wide range of studies, both in already cited by the authors (Lapa, et al. Vaccines 2022, 10, 817; Svetlova J, et al. Int J Mol Sci 2022, 23, 13220; Rodriguez, el al. Front Immunol 2022, 13, 992370; Gushchin, et al. Vaccines 2021, 9, 779) and in not cited (Bowen, et al. bioRxiv [Preprint] 2022, 2022.03.15.484542; Varese, et al. J Infect Dis 2022, 226, 1717-1720; Pascuale et al. Cell Rep Med 2022, 3, 100706; Gonzalez Lopez Ledesma, et al. mBio 2022, 13, e0344221; Komissarov, et al. J Immunol 2022, 208, 1139-1145; Sapkal, et al. J Travel Med 2022, 29, taac040). In this context, the key results of the study do not provide new data on this issue.
Nevertheless, the current study does possess a significant advantage over previously listed ones. In contrast to the present work none of the above mentioned studies analyses the antibody-dependent cellular phagocytosis of the SARS-CoV-2 variants analyzed. And this information is of a great value, since during the COVID-19 pandemic there were reported cases when the disease severity was elevated probably through the so called antibody-dependent enhancement (ADE) (see a good review: Sánchez-Zuno, et al. A review: Antibody-dependent enhancement in COVID-19: The not so friendly side of antibodies. Int J Immunopathol Pharmacol. 2021; 35:20587384211050199). In the context of ADE, the results obtained by the authors which demonstrated that antibody-dependent monocyte-mediated phagocytosis efficacy for D614G, B.1.617.2 and BA.1 variants did not differ significantly in vaccinated individuals, while the virus neutralizing activity were greatly reduced for BA.1, are new and represent a great value. These results may indicate that high IgG levels with low neutralizing activity against new SARS-CoV-2 variants may be harmful, and new vaccines which lead to the generation of the neutralizing antibodies are currently needed. Thus, I believe that the focus of the current paper should be transferred from neutralization assay to antibody-dependent cellular phagocytosis. This will greatly improve the paper and add a novelty to the results.
Additionally, I have several minor remarks which the authors should address.
1. Introduction: “…receptor-binding domain (RBD) located in S-1 fragment.” “S-1” should be replaced with “S1” without a dash.
2. Introduction: “…antibodies in both convalescent and vac-cine sera.” It is a typographical error, a dash in the “vaccine” should be deleted.
3. Introduction: “…in Sputnik-V vaccine sera neutralizing capacity [42; 24; 25] against Omicron variants…”. First, a dash in “Sputnik V” should be deleted. Second, the reference 42 deals with RBD-specific nanobodies from llamas and doesn’t contain any information concerning virus neutralizing activity of the Sputnik V sera. The reference should be removed.
4. 2.1. Ethics approval: “Protocol â„– 5 from 01/04/02/2021”. I believe that there are too many numbers for the date format. Please, check for typographical error.
5. Table 1. The study groups summary data. “Previously infected with COVID-19, n (%)”. Why do the authors sure that 62 vaccinated individuals were COVID-19 naïve? This disease has many non-symptomatic forms, and antibodies trend to decrease after the recovery, so even seronegative individuals may be not naïve. I recommend to use another term: “seronegative at a time of vaccination”. Additionally, why only “9 (35%)” individuals were previously infected with COVID-19 among SARS-CoV-2 recovered individuals? It seems that the number should be 26 (100%) since they are “recovered”.
6. 2.4. Plasmids. The authors must provide either complete sequences in Supplementary materials or database sequence IDs for ACE2, TMPRSS2 and 19-aminoacids-truncated S proteins used in the study.
7. 2.8. Antibody-dependent cellular phagocytosis. “…the biotinylated recombinant trimerized S-proteins of D614G, B.1.617.2 and BA.1 SARS-CoV-2 variants…”. Please, provide a supplier or describe how these proteins were generated.
8. 3. Results. “…collected in 1 month (n=36), 1-4 months (n-16) and…”. A dash in brackets should be replaced with “=” like “1-4 months (n=16)”.
9. The reference to Figure 1C is placed too far in the text, so I was a little confused while reading about the sera neutralizing activity changing dynamics, since it was not clear where these data were illustrated. I recommend to insert the reference to Figure 1C just in the end of the sentence: “For this we compared the sera samples from Sputnik V vaccinated individuals collected in 1 month (n=36), 1-4 months (n-16) and 4-6 months (n=10) after the second vaccine dose (Figure 1C).”
10. 3. Results. “We found that in 1 month after vaccination the sera neutralizing activity…”. I suppose that “in” can be deleted, just “We found that 1 month after…”.
11. Figure 1. Please indicate what parameter does the color scale represent in Figure 1B.
12. Figure 1C and 1D. Please indicate what do the gray zones represent.
13. 3.2. Previous Sputnik V vaccination does not enhance the sera neutralization activity against BA.1 for patients recovered from SARS-CoV-2 Omicron. Throughout the text the authors use the term “patients recovered from SARS-CoV-2 Omicron” or “Omicron recovered patients”. Please indicate if the viruses in these patients were isolated and characterized to be Omicron? If yes, please, indicate what method was used to characterize the SARS-CoV-2 variant. Otherwise, it is incorrect to use this term, since the authors indicated that participants’ recruitment lasted from May 2021 till March 2022 in Moscow, and, according to the published information (see Gushchin, et al. Dynamics of SARS-CoV-2 Major Genetic Lineages in Moscow in the Context of Vaccine Prophylaxis. Int J Mol Sci. 2022, 23(23):14670) during this period Delta and Omicron variants predominated. Therefore, these patients could be as well Delta or Delta/Omicron recovered.
14. Discussion. “…we carried out an independent assessment of Sputnik V vaccine sera neutralizing activity on a representative sample of a larger size…”. Please indicate what stands for “representative sample”. What do the samples collected by the authors represent – vaccinated population, Moscow population, Russian population? What statistical criteria and methods were used in order to ensure that the cohort enrolled in the study is representative?
15. Study Strengths and Limitations. “The main strength of this study is the use of high-throughput quantitative method of neutralizing antibody levels assessment without the use of infectious virus”. Indeed, the use of pseudoviruses is more safety compared to native viruses. However, this point is not only the strength of the current study, but also is its limitation. The surface of the real virus is a complex system, so the neutralizing effect of a serum can be different between real virus and S-protein bearing pseudovirus. This issue should be addressed in additional study. Please, discuss this issue in the limitations of the study.
16. Figure S1. Please indicate in the Figure S1 what does the dotted line reflect.
Reviewer 2 Report
The authors report neutralization and functional (ADMP) responses from individuals receiving the Sputnik V vaccine. Importantly, they explore responses from vaccinated only individuals as well as responses from convalescent individuals who then received the Sputnik V vaccine.
The manuscript is reads well, however most of the figures and figure legend lack clarity as to the cohorts are being shown.
Table 1 and Methods: The data for the SARS-CoV-2 recovered individuals is unclear. In the methods, it is mentioned that "In the recovered group, the serum samples were collected from individuals recovered from COVID-19 from December 2021- March 2022 (n = 26)." It is unclear from this or Table 1 if these individuals had the vaccines before or after their infection, or if they had multiple COVID-19 infections. Some clarity on the cohort is mentioned on Page 7 " For this we analyzed the neutralization activity of convalescent sera obtained from COVID-19 recovered patients (n = 26), among which 18 patients (64%) had been previously vaccinated with two doses of Sputnik V." However, this does not match the information list in Table 1 either.
Figure 1: It might be more beneficial to the reader if Figure 1A is moved to Supps, and instead figures for ADMP are moved to the main Figures instead. While there is mention of increased RBD mutations with BA.1, this manuscript is not focused on the individual mutations. As such, having Figure 1A visually represented in Supps would be suffice, and that frees up space on the main figure for ADMP data (which is currently only represented by the heat map).
Figure S1: It would be beneficial to the reader if these responses are tabulated by timepoints are vaccination, instead of having all 3 timepoints in the same graph. Having 1 graph for 1 month, another for 3 month, and lastly one for 6 months after vaccination would better show the decay or retention of responses over time (which is currently unclear from Figure S1).
Figure S2, Figure 1D: The figure shows n=18 vaccinated individuals, and n=8 unvaccinated individuals. However, these numbers do not match the details listed in Table 1 (previously vaccinated n=14, previously infected with COVID-19 n=9). It would be beneficial to clarify if these vaccinated individuals were infected then vaccinated, or vaccinated then infected, and if they received 1 or 2 vaccine doses. As that would skew the interpretations made in comparison to that for the vaccinated only cohort.
Figure S4 and S5: It is unclear what type of vaccine sera is used for these comparisons. Are these sera only from the vaccinated only individuals (n=62)? Is this from a single timepoint after vaccination? (1 month?, 3 month?). If COVID-19 recovered +vaccinated individuals are included in these figures, the results are even harder to interpret as individuals with variant infections could develop a broader antibody response as compared to wildtype vaccination. The number of antigen exposures, from the combination of infection and vaccination (1-2 infections, 1-2 vaccines?) could also be different from the vaccinated only cohort (2 vaccines)
The authors do show the ID50 responses for the respective timepoints for the vaccinated only cohort and COVID-19 convalescent cohort (Figure 1C, D). However this is not done for ADMP responses, and if present, this data could be moved to the main figure to aid interpretation of the heat map (Figure 1B)
Clarity around these aspects would greatly help the interpretation of data in this manuscript.
Stats: Figure 1D, S4, S5: It is unclear why Mann-Whitney paired analysis was used instead of Kruskal-Wallis for the grouped analysis (as seen in Figure 1C).
If the purpose for Mann-Whitney is for paired comparisons between vaccinated and unvaccinated cohorts in Figure 1D, it would be better to have 2 different graphs instead. One could remain as per 1D with comparisons made using Kruskal-Wallis within groups. The other is just paired graphs, in which BA.1 and D614G respectively are compared between vaccinated and unvaccinated cohorts.
Discussion: The authors propose that the lack of difference in ADMP responses could because the ADMP is not limited by neutralizing antibodies binding to the RBD. Indeed, for the phagocytosis assay, the authors used trimeric S-protein, which would account for the entire spike, not just the RBD. It should be noted that other groups (such as Bartsch et al 2022, Science Trans Med; Richardson et al 2022, Cell Rep Med) have seen differences in Fc-mediated responses against SARS-CoV-2 variants, depending on the variants tested and exposed to (from infection). The lack of differences observed in Figure S5 could be due to the lack of samples, or the aggregation of different cohorts (unclear if these are vaccinated only responses).
Reviewer 3 Report
Specifically
1. In table 1, for the samples from the recovered individuals, 14 have been vaccinated, and 9 have been previously infected. Any overlap for these two sub-groups. Could you distinguish them and analyze them specifically?
2. No correlation was found between Nab and ADMP responses. The authors proposed that the overall anti-SARS-CoV-2 antibody levels might related to the ADMP. The anti-N, S1 and RBD have been detected this communication. Could the author investigated the relationships of these markers with ADMP and update the discussion correspondingly.
After minor editing, the quality of the English language meets the criteria for publication.
Round 2
Reviewer 1 Report
First of all, I would like to appreciate the authors for the thorough addressing of all my comments. The manuscript has been greatly improved and now is acceptable for publication. Only a few minor points should be considered:
1. “2.2. Serum samples and study design. In the vaccinated group…” Please, indicate clearly what serology test was used. Since these participants had been vaccinated and obviously developed the antibody response against S-protein, I believe that the test was for detecting anti-N-protein specific IgGs. So, it should be stated like: “In the vaccinated group, the serum samples were collected from individuals (n = 36) who received two doses of Sputnik V vaccine and at the time of sample collection did not have confirmed SARS-CoV-2 infection, as well were seronegative for the anti-N-protein specific IgGs (Table 1).”
2. Lines 77-78: “Upon the sample collection, a PCR, analysis was performed, which allows to distinguish…”. This sentence is a little confusing. If the PCR analysis was performed at the same moment as the sample collection, this indicate that the participants were infected at the very moment of sample collection. Is it right? Please, describe more detailed this issue. Nine participants who “had previous COVID-19 infection according to their anamnesis data” – were their viruses characterized too using the same kit?
3. By the way – what type of samples were collected for the PCR analysis of the virus type – nasopharyngeal/nasal swab, saliva or else? Please, briefly describe the protocol of PCR analyses using AmpliTest, SARS-CoV-2 VOC v3, CV017, Russia, because quick search on the Internet provided no information concerning the kit. Please, provide a representative result for Omicron in the supplementary materials.
4. In Table 2 the row “Previously unvaccinated, n (%) 8 (31%)” is redundant and should be deleted. The authors have already provided the information about previously vaccinated participants (18 (69%)). It is obvious, that the rest of them are non-vaccinated.
5. I am totally confused. According to the text in lines 77-79, all participants from the “recovered group” were tested for SARS-CoV-2 using the AmpliTest kit, and the presence of Omicron in the samples was confirmed. Why then in Table 2 only 9 (35%) of participants had confirmed cases of COVID-19? Please, clarify this moment.
